# Predictability of fossil fuel $CO_2$ from air quality emissions

**Kazuyuki Miyazaki** [⦿][1] [✉] **& Kevin Bowman** [⦿][1]

Quantifying the coevolution of greenhouse gases and air quality pollutants can provide insight into underlying anthropogenic processes enabling predictions of their emission trajectories. Here, we classify the dynamics of historic emissions in terms of a modified Environmental Kuznets Curve (MEKC), which postulates the coevolution of fossil fuel $CO_2$ (FFCO2) and NOx emissions as a function of macroeconomic development. The MEKC broadly captures the historic $FFCO_2$-$NO_x$ dynamical regimes for countries including the US, China, and India as well as IPCC scenarios. Given these dynamics, we find the predictive skill of FFCO2 given $NO_x$ emissions constrained by satellite data is less than 2% error at one-year lags for many countries and less than 10% for 4-year lags. The proposed framework in conjunction with an increasing satellite constellation provides valuable guidance to near-term emission scenario development and evaluation at time-scales relevant to international assessments such as the Global Stocktake.

Fossil fuel $CO_2$ (FFCO2) emissions continue to be the largest driver of anthropogenic climate change[1] while co-emitted air pollutants are one of the largest global morbidity factors[2]. Broadly speaking, both emissions are driven by common activity such as fuel consumption, but differ by their relative contribution (i.e., emission factor). At country scales, fuel consumption is regulated by macroeconomic processes reflected in metrics such as gross domestic product (GDP) while emission factors reflect sector distribution, technology, and environmental regulation. A framework for understanding the balance between activity and emission factor during economic growth is the environmental Kuznets curve (EKC), which hypothesizes that at early stages there is a regime where economic development (activity) is prioritized at the expense of environmental quality (factor) but transitions over time to a regime where additional resources are placed on limiting environmental degradation (i.e., reduced emission factors) reflecting changes in both sectoral distribution and economic costs. The EKC leads to an inverted "U" shape curve of environmental degradation as a function of GDP[3]. The EKC concept has been shown to be useful for interpreting IPCC scenarios[4] and has been tested for many countries, including the United States (US)[5], Asia[6], India[7], and Africa[8]. For instance, in the US, consumption-based post-trade EKCs peak at significantly higher incomes than production-based pre-trade EKCs, suggesting that emissions-intensive trade largely drives the income-emissions relationship[9]. The EKC can also be applied to sub-national scales. For example, per capita GDP appears to be the primary driver for $FFCO_2$ from Chinese cities, indicating that the economic wealth of cities may be a key factor in driving their emission changes[10].

Accurate and timely $FFCO_2$ emission inventories are critical inputs for assessments of the global carbon cycle[11], climate targets[1] as well as for carbon cycle assimilation. However, these inventories are dependent on country reporting, which can take several years to produce[12]. While these emissions are well-known relative to other parts of the carbon cycle, inventory compilation can still incur significant regionally-dependent uncertainties[13,14], which are typically 5–10% for developed countries but likely higher for developing countries[15]. These differences can confound the attribution of natural and fossil fuel trends[16]. For example, large variations in $FFCO_2$ in China among nine inventories were largely due to the different emission factors and activity data[17]. Spatially-explicit inventories depend on proxies such as population and remote sensing in order to spatially allocate country totals[18–21]. Differences in these approaches can lead to large discrepancies in spatial patterns at city-scales[22–24].

Air quality (AQ) emission inventories, like $FFCO_2$, use similar methods to determine fuel consumption and sector-based emission factors, and consequently incur substantial latency in their reporting[25].

[1]Jet Propulsion Laboratory, California Institute of Technology, Pasadena, CA, USA. [✉]e-mail: kazuyuki.miyazaki@jpl.nasa.gov

However, recent advances in satellite-based AQ emissions from chemical data assimilation ("top-down") estimates can circumvent challenges with latency while providing an unprecedented picture of global AQ emissions. These systems have been applied to both short-term (e.g., during the COVID-19 lockdown)[26,27] and long-term global changes[28,29] in $NO_x$ emissions—one of the key atmospheric precursor pollutants—while accounting for chemical and transport processes[30]. Atmospheric data are a direct constraint on the product of activity and emission factor and therefore an independent dataset for inventory evaluation.

Information from AQ measurements can provide supplemental information on $FFCO_2$ estimations. Proxy species such as $NO_2$ and CO can help monitor localized $FFCO_2$ due to their relatively short lifetime and large signal relative to $CO_2$[31–33]. At large scales, however, $FFCO_2$ and AQ emissions will evolve as a consequence of increased regulation, changes in sector composition, and economic development[34,35]. Consequently, the coevolution of $FFCO_2$ and AQ emissions can provide insight into the underlying anthropogenic processes. This insight, in turn, could enable the predictability of emissions trajectories at national and subnational scales. This information is increasingly urgent to support short-term climate mitigation assessments such as the Global Stocktake needed for the Paris Agreement and international efforts[36], and how they may be coupled to AQ human health and vegetation impact assessment including both local and remote impacts[37].

In this study, we quantify the co-evolution of $FFCO_2$ and $NO_x$ emissions over the last two decades using a combination of bottom-up $FFCO_2$ and top-down AQ emissions. We propose a modified EKC (MEKC) framework to classify this co-evolution, which are shown to be distinct for countries such as the US, China, and India. We then build a simple model to assess the predictability of $FFCO_2$ given low-latency $NO_x$ emissions. The performance of this system is assessed for a number of countries and is shown to be related to their MEKC regime.

## Results

### Modified EKC for $FFCO_2$ and $NO_x$

The original EKC describes environmental degradation as a function of economic growth[38], which implicitly assumes a monotonic increase in GDP over time. This framework posits an inflection point at some economic threshold whereby this degradation begins to diminish as environmental regulations are implemented and sector composition

shifts to lower AQ emissions. While both AQ and greenhouse gas (GHG) impact the environment, the time-scales between the two can differ substantially. Here, in order to evaluate the co-evolution of AQ and GHG emissions, we propose a modified EKC (MEKC) that describes the evolution of $FFCO_2$ and $NO_x$ emissions as a function of GDP. In this MEKC framework, GHG and AQ emissions are normalized to a reference year, leading to a spiral form (Fig. 1). Different regimes of GHG-AQ coevolution are represented on this form, denoted as quadrants (Q1-Q4). Here we use $FFCO_2$ and $NO_x$ as one of the most measurable GHG and AQ species and good proxies of various co-emission sources to describe the MEKC trajectory. MEKC applications to other forms of co-emitted GHG-AQ species would provide different trajectories and unique insights into the economy and emission relationship. Nevertheless, its concept provides a generalized framework to describe the emission trajectory dynamics.

At the initial stages of economic growth both $FFCO_2$ and AQ emissions trajectories follow a "business-as-usual" (BAU) phase (Q1) where $FFCO_2$ is driven by fuel consumption from increased GDP[39] and AQ emissions are weakly regulated. As economic growth continues, AQ emissions will start to decrease due to AQ mitigation and sectoral shifts while $FFCO_2$ continues to increase (Q2). In this case, AQ emissions can return to reference values or below (e.g., AQ emissions < 1). As the economy continues to mature, both $FFCO_2$ and AQ emissions decrease where $FFCO_2$ will return to its initial values while AQ emissions will be likely less than the normalized year (Q3). Finally, $FFCO_2$ emissions continue to fall below reference values as the economy moves towards decarbonization along with AQ emissions (Q4). This phase reflects substantial changes in energy production and AQ mitigation. In all phases, GPD continues to increase. Implicit in the MEKC formulation is the lag in the reduction of $FFCO_2$ relative to AQ emissions. The formulation assumes that an economy will address short-term health needs before long-term climate concerns, while achieving co-benefits between AQ and climate[40,41]. How well this assumption holds can be tested with predictability of the system.

If AQ and GHG evolve according to the MECK, then we could expect some predictive skill when the emission dynamics are in a particular regime (i.e., quadrant). Within the regime, the $CO_2/NO_x$ emission ratio should vary more slowly than either emission independently because the sectoral distribution should stay relatively stable given a common activity level.

To evaluate the predictability, we implemented a simple Kalman filter (KF) of the $CO_2/NO_x$ emission ratio (Fig. 2a). The $FFCO_2$ emissions (Fig. 2c) are then updated based on the product of the $CO_2/NO_x$ ratio prediction (Fig. 2a) and the top-down $NO_x$ emission estimate (Fig. 2b). In principle, top-down $NO_x$ emissions constrained by satellite observations can be computed with a much shorter latency (e.g., weeks) than $FFCO_2$ bottom-up inventories. Traditional methodologies for

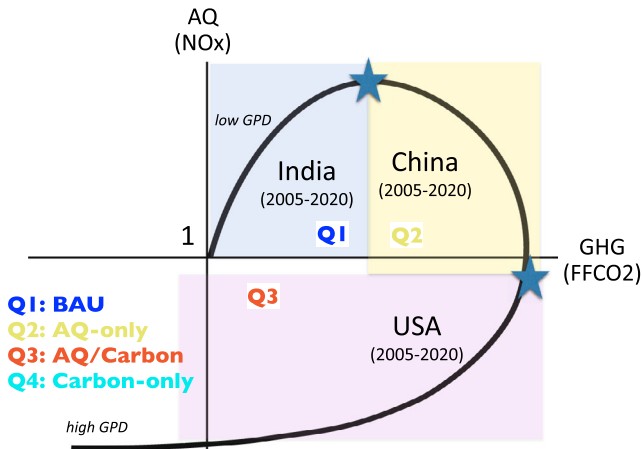

**Fig. 1 | The schematic diagram of the modified environmental Kuznets curve (MEKC).** The MEKC concept represents the relationship between greenhouse gas (GHG) emissions (x-axis) and air quality (AQ) emissions (y-axis) under developing economy. Q1 is the business-as-usual (BAU) condition. Q2 is the AQ only reduction phase. Q3 is the AQ and GHG co-reduction phase. Q4 is the Carbon-only reduction phase.

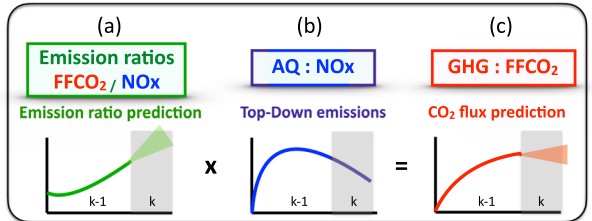

**Fig. 2 | The schematic diagram of the Kalman filter (KF) prediction. a** Changes in $CO_2/NO_x$ emission ratio for the previous time periods ($t = k - 1$) are diagnosed using top-down $NO_x$ emissions and bottom-up fossil fuel $CO_2$ ($FFCO_2$) inventories. **b** $CO_2/NO_x$ emission ratio for more recent time periods ($t = k$) is predicted using a KF optimal estimation approach based on information obtained from the previous time period ($t = k - 1$). **c** A prediction of $FFCO_2$ for $t = k$ is obtained from the predicted emission ratio from **b** and the top-down $NO_x$ emissions.

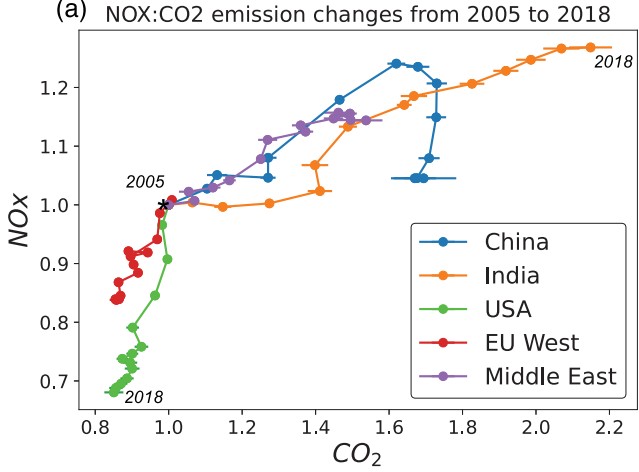

(a)  NOX:CO2 emission changes from 2005 to 2018

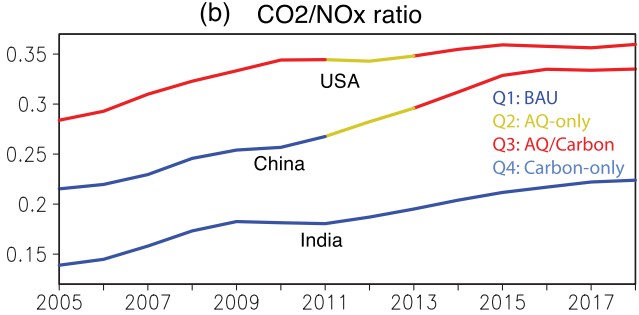

(b)   CO2/NOx ratio

**Fig. 3 | Co-evolution of anthropogenic emissions of CO₂ and NOₓ. a** Changes in country-total anthropogenic emissions of CO₂ (x-axis) and NOₓ (y-axis) from 2005 through 2018. The values normalized at the 2005 level are shown for China, India, USA, EU West, and Middle East. The NOₓ emissions were obtained from the TCR-2 top-down estimates. The CO₂ emissions were from ODIAC. The error bars represent one standard deviation of the CO₂ emission changes derived from the multi-inventory spread. **b** Time series of CO₂/NOₓ emission ratio from 2005 through 2018 for USA, China, and India. The modified environmental Kuznets curve (MEKC) phase is shown in different colors.

bottom-up inventories require country-scale reporting that can be delayed by several years. While utilizing sectoral distributions from bottom-up inventories informed by international statistics, our approach exploits both the rapid update of NOₓ emissions enabled by satellite assimilation and the gradual changes in technology and regulation (i.e., emission ratio). The approach is depicted in Fig. 2 and explained in the Material and Methods section.

We focus mainly on the country and annual scales because they provide the best data to evaluate the predictive skill within a MEKC regime. The KF prediction exploits the relatively gradual changes in emission ratio with a given MEKC regime. Even when the activity changes rapidly, multi-sector activities can be strongly linked at country scales and can provide robust KF predictions. However, during the transition from one regime to the next, we would expect poorer performance.

We complement the top-down NOₓ emissions with bottom-up FFCO₂, which use similar input data sources but employ different methods to spatially disaggregate country-level totals. The differences impact both national and subnational trajectories. We apply the KF to an ensemble of inventories to account for structural differences in the emission approaches on emission trajectory. Further detailed information is given in the Methods section.

## Co-evolution of FFCO₂ and NOₓ emissions

While the focus is on the recent two decades, a longer window provides context and insights into the dynamics described by the MEKC.

Using EDGAR historical inventories (Supplementary Fig. S1), the co-evolution of NOₓ and CO₂ emissions from 1970 to 2015 for the US show distinct dynamical patterns. From 1970-1982, the US experienced economic stagnation and concomitant GDP fluctuations leading to oscillations in CO₂ emissions but gradual reductions in NOₓ emissions from increased regulation. Renewed economic growth from 1982 onward led to relative increases in both NOₓ and CO₂ emissions, reflecting the phase Q1 of the MEKC (Fig. 1). At the turn of the century, AQ started improving but FFCO₂ remained stable, which suggests an inflection point indicative of a Q2 phase. After 2004, both FFCO₂ and NOₓ emissions started to decline reflective of a Q3 phase. Based on EDGAR sectoral information, this co-evolution is attributable to changes in energy usage, especially the reduction of coal, the shift towards natural gas, and the increase in renewable energy sources. Both China and India reveal strong long-term increases in country total emissions of both CO₂ and NOₓ from 1970, with a stronger increase in China until 2011, showing the continued GDP-driven economic development and environmental degradation consistent with phase Q1. Following the US, China moved to the phase Q2 after 2012.

The evolution of NOₓ and CO₂ emissions exhibit distinct dynamical patterns of GHG and AQ trajectory that can be interpreted in the MEKC framework for different countries during 2005–2018 (Fig. 3a) when top-down NOₓ emissions from chemical data assimilation (see Materials and Methods) are available. The NOₓ emissions have been extensively used to study decadal and short-term variability of anthropogenic emissions[26–28,42–44]. For China, both FFCO₂ and NOₓ emissions continued to increase until 2012 where the NOₓ emission trajectory reached a plateau, then exhibited an arc towards lower NOₓ emissions before returning to 2005 emission levels. FFCO₂ showed modest reductions after 2014. These changes show the MEKC phase shift from Q1 to Q2. A sectoral analysis based on the EDGAR inventories (Supplementary Fig. S2) suggests that the strong NOₓ emission reductions are driven by power industry sources (about 50%) followed by combustion for manufacturing (about 40%) from 2011 to 2018. For FFCO₂, while power industry has the largest contributions to the total increase, the relative growth rate was the largest for road transportation. Consequently, the phase shift from Q1 to Q2 was mainly driven by power industry and combustion for manufacturing. India demonstrated strong, monotonic increases in both NOₓ and FFCO₂ as well as GDP from 2005 to 2018, reflecting the BAU condition (Q1). The rate of increase is almost constant in NOₓ emissions from 2009 onwards and in FFCO₂ from 2005 onwards consistent with a Q1 phase. Based on the EDGAR inventory, about 78% of the increases are associated with power industry for NOₓ, whereas both power industry (60%) and combustion for manufacturing (24%) contributed to the FFCO₂ increase from 2011 to 2018. The different changes in emission ratio among sectors (Fig. 4) suggest an importance of sectoral-level emission ratio prediction to improve FFCO₂, as will be discussed later. A longer term analysis using the EDGAR inventories revealed that the BAU condition (Q1) continued for China from 1970 till 2012 before reaching Q2 whereas for India has persisted in Q1 from 1970 to the present (Supplementary Fig. S1).

Based on Fig. 3, US remained in the Q3 phase with AQ/Carbon co-reductions from 2005 to 2018. The US showed about a 20% reduction in NOₓ emissions in 2010 relative to 2005, followed by much smaller reductions in both FFCO₂ and NOₓ emissions thereafter, mainly driven by power industry emissions. These changes are attributable in part to road transportation reductions (24%), which contributed to a slightly increased FFCO₂ after 2010.

These countries show development at different points along the MECK trajectory. In many developing countries from 2005 to 2018, the MEKC phase shifted from Q1 to Q2. For example, after 2017 NOₓ emissions in Vietnam stay almost constant during 2017 and 2018, which are 20% higher than the 2005 levels, whereas FFCO₂ continues to increase to about 65% higher in 2018 than in 2005 (not shown).

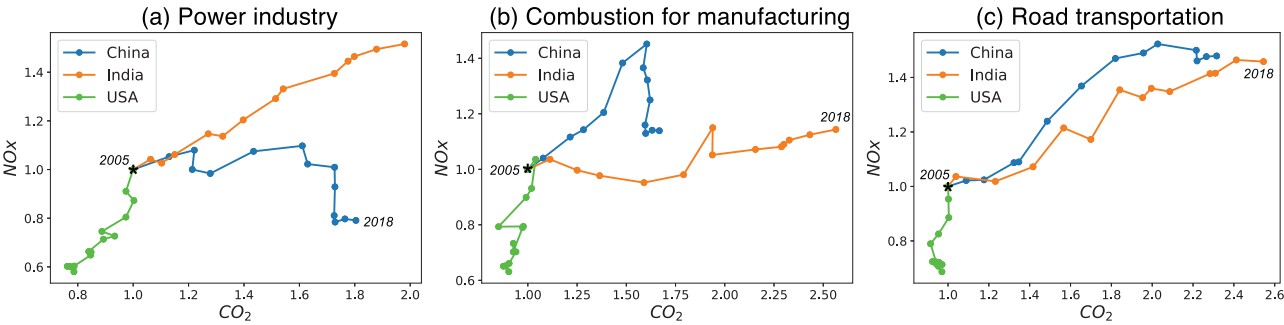

**Fig. 4 | Co-evolution of sectoral emissions of $CO_2$ and $NO_x$.** Same as Fig. 3a, but for $NO_x/CO_2$ emission changes for each emission sector separately: (**a**) power industry, (**b**) combustion for manufacturing, and (**c**) road transportation.

Similarly, the $NO_x$ emissions in Iran stay almost constant from 2013 to 2018, about 25% higher than the 2005 level, whereas $FFCO_2$ shows a nearly linear increase from 2005 through 2018 by 45%. In contrast, some developed countries such as Japan and western European countries have experienced gradual changes from Q3 to Q4. Non-monotonic changes in economic activity can obscure the underlying trajectory for the MEKC, such as the 2007–2008 global financial crisis with widespread negative emission anomalies for both $FFCO_2$ and $NO_x$ across the US, Europe, and many other countries.

The $CO_2/NO_x$ emission ratio is higher in 2018 relative to 2005 for the US, China, and India (Fig. 3b), which reflects a gradual decoupling of AQ emissions from energy production possibly reflecting the adoption of clean air technologies and a shift in sectoral composition[25]. The $CO_2/NO_x$ ratio for the US has stabilized to 0.35 PgC/TgN during the time period. The sectoral emission ratios for the US were higher in power sector (0.42–0.58 PgC/TgN) than the combustion (0.22–0.3 PgC/TgN) and transportation sector (0.15–0.2 PgC/TgN), while the flattened ratio trend in 2016–2018 is largely driven by the transportation and power sectors (Supplementary Fig. S3). The current estimate of about 0.35 PgC/TgN is consistent with other developed countries such as Europe and Japan. This plateau could indicate that AQ reductions are nearly saturated in the Q3 phase. By contrast, in most developing countries including Vietnam, Iran (not shown), and China, the emission ratio continues to increase. The rapid increase in China's emission ratio until 2016 and flattened trends afterward are driven largely by power sectors, followed by transportation. However, the difference in the emission ratio between the US and China has narrowed from 0.07 TgC/TgN to less than 0.03 TgC/TgN from 2005 to 2018. India's ratio has improved by about 20% but is still substantially less than either China or the US.

The GHG-AQ emission ratio tends to increase with GDP, demonstrating the coupling between economic development and technology improvement informed by AQ and carbon mitigation (Supplementary Fig. S3). There are differences between the emission ratio for countries at the same GDP. For example, India's GDP in 2018 is about the same as China's in 2005 (approximately 2.5 trillion US$) as are their emission ratios, population (1.30 billion in China in 2005 and 1.35 billion in India in 2018) and per capita GDP. On the other hand, GDP in China in 2018 is close to the US in 2005 (roughly 1.35 trillion US$), but the emission ratio is higher for China in 2018 than the US in 2005. For example, the emission ratio in China's power sector in 2018 is about 0.57 TgC/TgN, which is comparable to the US emission ratio of 0.59 TgC/TgN, but is substantially higher than the US emission ratio in 2005 (0.42 TgC/TgN) for the same sector (Supplementary Fig. S3). The parity in emission ratio suggests that a country can adopt modern industrial technology allowing them to accelerate its MEKC trajectory.

The choice of inventories affects the decadal emission trajectories (Supplementary Fig. S4). The trend in sign for India, China, and the US are consistent between inventories. Nevertheless, the magnitude of $FFCO_2$ differs by up to about 10% at country scale, with even larger differences at smaller scales. Our top-down $NO_x$ emissions provide unique information on the trajectories, such as smooth gradual changes for India during 2005–2018 and flat emissions since 2016 for China, but tend to be smoother than the EDGAR emissions (Supplementary Fig. S4). Because of strong observational constraints by assimilated satellite measurements, the choice of prior emissions has a reduced influence on the optimized $NO_x$ emissions[42,45]. Consequently, top-down $NO_x$ emissions represent both a potential benchmark for bottom-up estimates and a way to reduce latency for the recent past, while providing improved estimates, especially in regions where global inventories lack accurate and timely activity data and emission factors[43].

**Prediction based on multiple emission inventories**

The distinct changes in the MEKC trajectory provide insights into not only emission processes but also the skill in predicting $FFCO_2$ for different regimes. The relatively smooth changes in the GHG-AQ emission ratio suggest that the evolution can be predicted within MECK regimes. Figure 5 shows the time series of $FFCO_2$ estimated from the prediction of the $CO_2/NO_x$ emission ratio and the top-down $NO_x$ emissions. Emission ratios are trained on individual bottom-up inventories from 2005 to 2015 and then those ratios are predicted and applied to estimate $FFCO_2$ for 2016–2018. The prediction error for 2016–2018 is computed relative to each withheld inventory. The spread among four different $FFCO_2$ inventories is used as a proxy for the uncertainty in country total emissions. This spread is compared to the range of predicted $FFCO_2$ across inventories. As expected, prediction error increases with time as the emission ratio evolves relative to the 2015 value.

Even at country scales there are substantial differences between inventories even though global $FFCO_2$ agree well. For instance, the ODIAC and EDGAR showed minor differences in magnitude (0.3–2.7%) and trends in global total emissions[18]. At country scale, China's $FFCO_2$ varies by almost 30% from about 2.1 to 2.8 PgC in 2015. These differences reflect different regional activity data, emission factors, and latency of data during inventory compilation. The comparisons in Fig. 5 also highlight that the temporal evolution of $FFCO_2$ is strongly dependent on the bottom-up approach. The difference in sectoral definition, resolution, and methodology can also result in the multi-inventory discrepancy (see Materials and Methods).

In order to separate the performance of the emission ratio prediction from the spread of the inventories, all inventories were shifted to the same 2015 value (e.g., 2.8 PgC for China). The adjusted inventories, without applying the KF prediction, show different temporal evolution during 2016–2018, with the multi-model spread (1-$\sigma$) of up to 2.1% for China and 3.3% for the US in 2018. The ODIAC inventory exhibits substantially lower $FFCO_2$ in 2017-2018 in the US compared to the EDGAR and GCP inventories, reflecting different trends from 2016. The multi-inventory agreement is better for India, with up to a 1% spread using the shifted emissions.

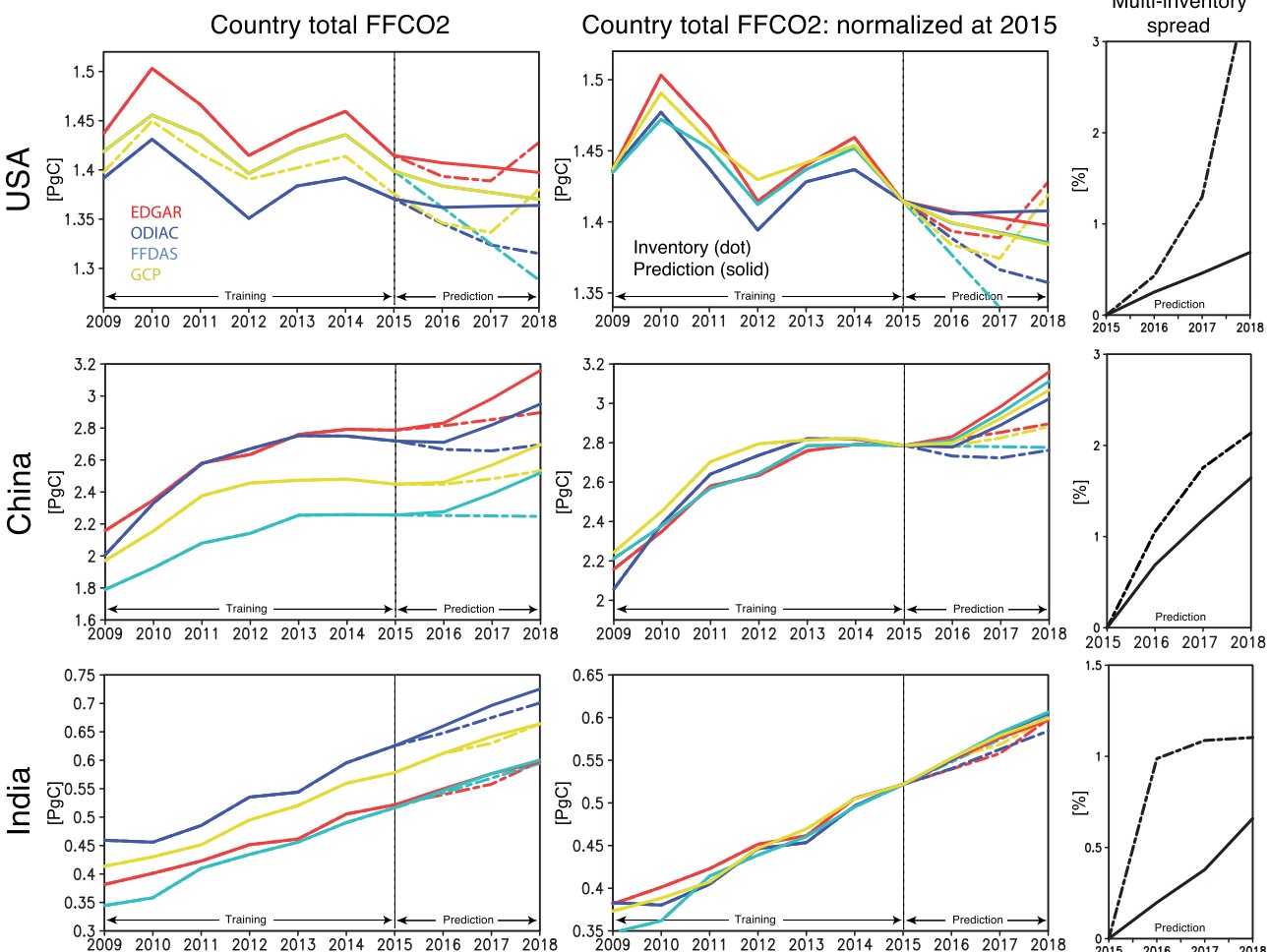

**Fig. 5 | Temporal evolution of fossil fuel CO₂ (FFCO₂).** (Left) Time series of country total fossil fuel CO₂ (FFCO₂) obtained from multiple inventories: EDGAR (red), ODIAC (blue), FFDAS (magenta), and GCP (orange) during 2009 and 2018. The original inventory values are shown by dotted lines. The Kalman filter (KF) prediction results, with training before 2015, are shown by solid lines. (Center) Similar to the left panels, but the emissions inventories shifted to a common value (to match the EDGAR emission value) at 2015 were used to make the predictions. The shifting was applied to avoid the influences of systematic differences among the inventories on the KF predictions, (Right) The multi-inventory spread after 2015 for the original inventory (dotted lines) and KF predictions (solid lines) using the shifted inventories. The results are shown for China (top), USA (middle), and India (bottom).

The KF predictions reduced the spread between the inventories in 2016–2018, reflecting the common constraints provided by the top-down NO$_x$ emissions. In the case with applying the 2015 normalization, the multi-inventory spread is reduced by 80% over the USA in 2018, from 3.3% spread in the original inventories to 0.6%. The multi-inventory spread is also reduced over China by 25% and over India by 45% in 2018. These results suggest that the common NO$_x$ emissions estimate provides a more consistent calculation of activity while the KF ratio prediction provides a more consistent model of temporal dynamics than implied by the bottom-up inventories. Consequently, these results lead to a more precise estimate than from the multi-inventory.

The country-scale results in Fig. 5 are based on a 3-year prediction window, which is a typical latency for comprehensive bottom-up inventories while there has been an increasing attempt to reduce its latency up to a year using relatively simplified settings. The predictability of the emission ratio depends not only on the lag-time but also the regional emission dynamics. Figure 6 shows one-year KF predictions initialized for each year from 2005 to 2018 trained against ODIAC inventories. The KF captures the changes in emission ratios within MEKC regimes. However, the approach does not reproduce the dynamics when the emission ratio changes rapidly, such as in India in 2010 and the USA in 2007 during the economic crisis. These anomalies could not impact all sectors equally, which leads to a change in the aggregate emission ratio, and therefore degrades the prediction skill, especially when predicting FFCO₂ at small scales where the relative sectoral distribution can change substantially, e.g., transportation relative to power production. At country scales, however, multi-sector activities are highly coupled and therefore provide robust predictive still for many cases, as discussed later.

The prediction error was calculated from differences between the predicted and original inventory values (Fig. 6d). When the emission ratio changes are temporally smooth, the KF prediction error is generally less than 2% errors for both developing (e.g. Vietnam and Iran, not shown) and developed countries (Fig. 6d). The exception is India in 2007 and 2010 where the 1-year lag error exceeded 3%. Likewise, US prediction overestimated FFCO₂ reductions from 2006 to 2007 related to the economic crisis (Fig. 5c). Short-term fluctuations in GDP are not well-modeled in the MEKC and are reflected in the skill of the prediction. In general, however, these errors are smaller than the spread in current emission inventories (6–7%)[14]. Rapid changes in emissions are often driven by changes in activity that are well-informed by satellite-constrained NO$_x$ emissions, e.g., COVID-19 lockdowns[27,44] and cut across multiple sectors. To the extent that the relative sectoral impacts are the same, the FFCO₂ will be robust. Over longer time

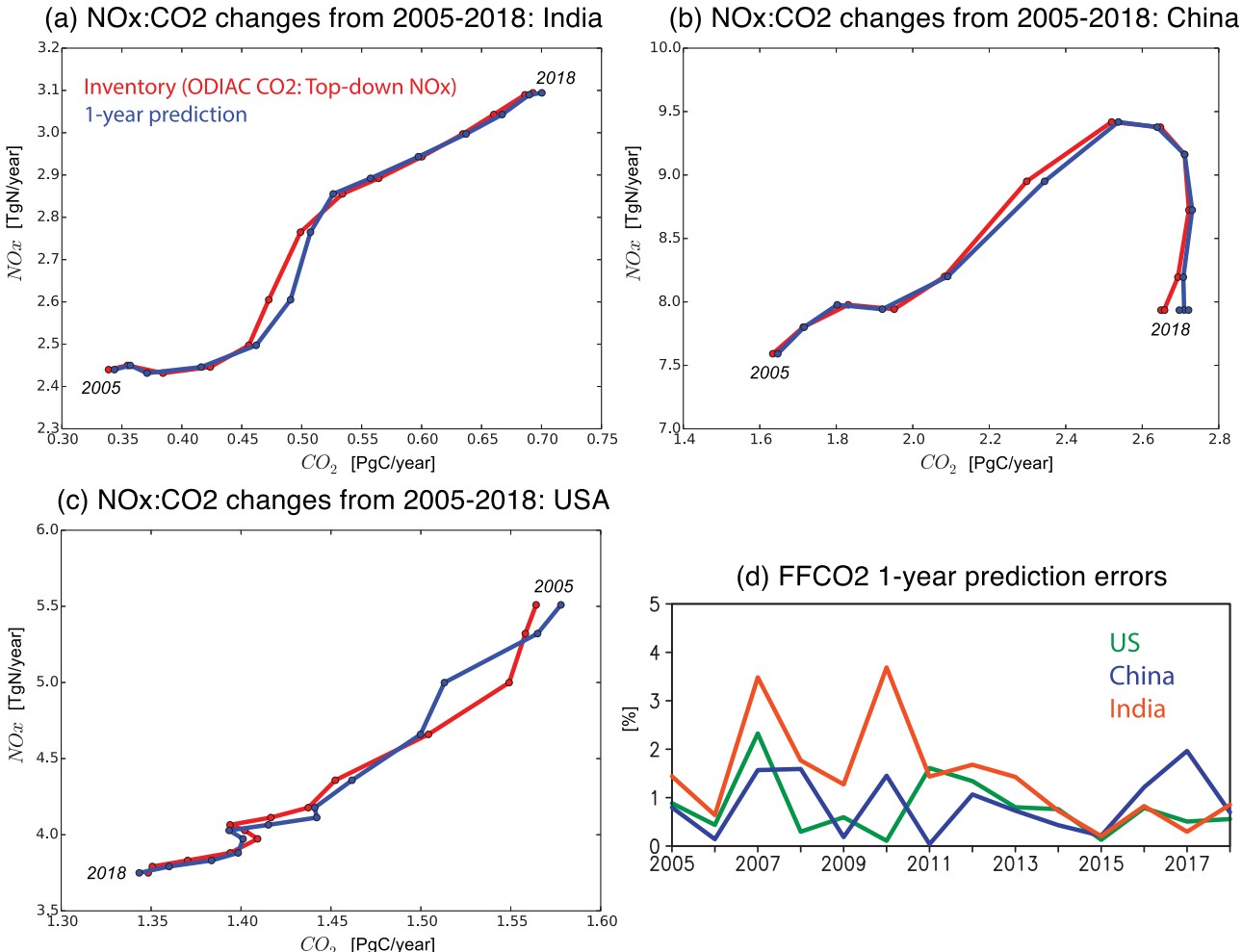

**Fig. 6 | One-year Kalman filter (KF) prediction performance. a–c** Co-evolution of fossil fuel $CO_2$ (FFCO$_2$) from the ODIAC inventory and NO$_x$ emissions from top-down estimates (red). The results obtained from the one-year Kalman filter (KF) prediction are also shown (blue) for (**a**) India, (**b**) China, and (**c**) US. **d** Time series of root-mean-square (RMS) errors of the one-year KF FFCO$_2$ predictions (in %) for US, China, and India.

scales, the predictive skill suggests that emission ratios tend to change more slowly than activity.

The limit to useful predictive skills was investigated by initializing each year from 2010 to 2017 (Fig. 7). The growth rate of prediction error depends largely on the start year of the KF prediction. For example, in the US, the prediction error did not exceed 2% for 1 to 4-year lags starting from 2011 and 2013. Starting from 2010, however, the 4-year lag error exceeded 5%. However, for other years (7 out of 8 years), the 4-year lag error did not exceed 5% while all the 1-year lag predictions had errors less than 2%. China and India show similar predictive skills, with slightly smaller errors for China than India for 1–4 year lag predictions. The mean 2-year lag errors are about 2% for China and 3% for India, whereas both countries reveal about 5% mean errors for the 3-year lag predictions and 7% for the 4-year lag predictions. The 4-year lag errors exceed 10% only for China starting from 2014 and in India starting from 2011. The relatively large errors in China starting in 2011 are reflective of the regime shift in AQ-Carbon starting in 2011. The high predictive skills for India reflect the continued linear increase in both emissions consistent with Q1 dynamics and substantial increases in GDP.

As discussed in the Materials and Methods section, the uncertainty estimate is robust based upon three independent uncertainty estimates: (1) KF prediction errors against the original bottom-up inventories, (2) multi-inventory spreads of the predicted FFCO$_2$, and (3) predicted FFCO$_2$ uncertainty from the KF equations.

## Sectoral analysis

Emission ratios differ between regions as a consequence of sectors and their relative activity (see Methods). To understand their impact in greater detail, we investigated the sectoral drivers using the EDGAR sector-specific grid map[25] that could provide insight into emission processes.

Based on the EDGAR inventory in 2018 (not shown), the power industry accounted for about 38% of total FFCO$_2$ in the US, followed by road transportation (30%) and energy from buildings (11%). In both China and India, the power industry has greater contributions (44% in both countries) than the US, with the second largest contribution from manufacturing (29% and 22%, respectively). As shown in Supplementary Fig. S2, the MEKC phase change from Q1 to Q2 in China is largely driven by manufacturing and power industry sources, which largely dominate over transportation sources. In India, the relative distribution of sectors remains stable though manufacturing has the greatest contribution to the ratio increase (figure not shown).

The emission ratios show distinctly different patterns among sectors (Fig. 4). For instance, in China and the US, the emission ratio of the power industry emission increased due to the AQ regulations and the increased use of natural gas[46]. Also, in the US, total on-road NO$_x$ emissions declined after 2004 when heavy-duty diesel NO$_x$ emission controls started[47], which increased the emission ratio.

The MEKC framework is robust for interpreting GHG-AQ co-evolution when integrated over coupled sectors typical of countries

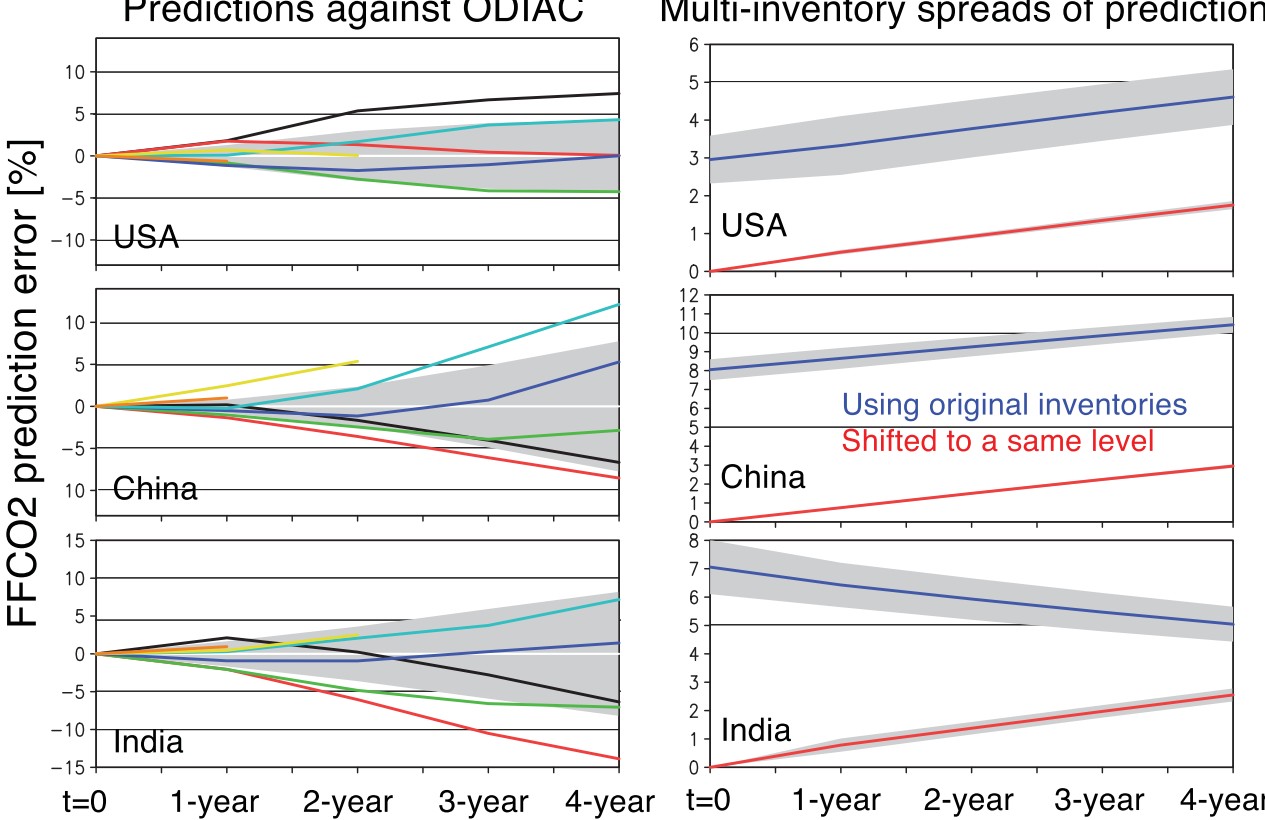

**Fig. 7 | Temporal evolution of the Kalman filter (KF) prediction errors.** (left) Temporal evolution of the Kalman filter (KF) prediction errors, starting from each year during 2010 and 2017 (color lines), for the US, China, and India. The shaded area represents the ± root-mean-square (RMS) of the prediction errors. (right) The

multi-inventory spread of KF predictions using the original inventories (blue) and inventories shifted to a same level at the beginning of the prediction ($t = 0$). The shaded area represents the 1-sigma deviations of the spread among predictions staring in different years (2010–2015).

scales. However, individual sectors may deviate from the MEKC. For example, FFCO2 from European transportation has increased since 2013 while NOx emissions decline due to the growing demand for freight transport and the effective AQ regulation for heavy-duty vehicles. That sector change is more reflective of Q2 even though Western Europe as a whole is in Q3 where both $CO_2$ and $NO_x$ emissions have reduced. At regional scales, the ratio of aggregated sectoral $CO_2$ emissions to aggregated sectoral $NO_x$ emissions is not equal to emission ratios aggregated over sectors (see Methods). Consequently, aggregated over all the major sectors, countries such as US, India, and China, and western Europe follow the MEKC regimes, but individual sector emission ratio trajectories may have distinctly different trends.

Nevertheless, our comparisons show that projecting KF predictions to sectoral scales were able to provide predictive skill for most dominant sectors. The predictive errors of total $FFCO_2$ were mostly comparable between the cases with and without sectoral information (see the Methods section) with less than 5% difference in the predictive errors for the US, China, and India at country scale throughout the analysis period. However, using emissions at each grid point, the predictive errors became slightly larger when sectoral information is used, by about 0-60 % (from about 0.5–3% to 0.5–4%) for 1-year prediction for China and by up to 100% (from 0.5–2% to 0.5–4%) for India (figure not shown). The difference was smaller for the USA. The overall increased error could reflect a more obvious transition in the MEKC regime for the individual sector compared to those in total emissions at the individual grid point. The comparisons also highlight that the impact of the sectoral shifts informed by bottom-up inventories is well reflected as a whole in changes in an aggregated country total emission ratio. An aggregate emission ratio usually shows smooth trajectories (Supplementary Fig. S3). Since a KF prediction error of total emissions

can be represented as a sum of sectoral emission ratio prediction errors (see the Methods section), a smooth trajectory of an aggregate emission ratio is more amenable to KF predictions.

The estimated emission trajectories for each sector separately can be used to identify key processes, resulting in changes in the emission dynamics. For instance, in the US, the KF prediction starting in 2011 showed a decreased relative contribution from power industry from 42% to 38% in 2015, similar to the original inventory for the same time period (from 42% in 2011 to 39% in 2015), while suggesting about a 4% increase in $CO_2/NO_x$ ratio for that sector. In contrast, the KF predicted the increased relative contribution from power industry, from 42% in 2011 to 47% in 2015, again consistent with the original EDGAR inventory.

### Regional scale analysis

The relationship between emissions and GDP for subnational scales has been described by the traditional EKC. For example, peaks of per capita emissions and the years that these peaks occurred differ significantly across many Chinese cities[10], but these changes are expressed differently among the $FFCO_2$ inventories. For rapidly developing cities in Asia[20] and Middle East[48], strong increases in both AQ and GPD are attributable to local economic developments.

Consequently, the relationship between AQ and GHG emissions could also be well-described even at subnational scales. Nevertheless, the KF prediction skill is scale-dependent (Supplementary Fig. S5). The prediction error generally increases with increasing spatial resolution (as does a priori uncertainty). As shown in Supplementary Fig. S5, the 1-year KF prediction error strongly depends on the KF prediction resolution. At $0.1 \times 0.1°$ resolution, it exceeds 10% for about 14% of the grids, which is only about a factor of 2 worse than the global mean

1-year predictive error of 5.5% over high emission grids ($FFCO_2$ greater than 0.1 gC/m2/day).

The high-resolution prediction provides detailed information on spatial gradients in the emission trajectories. $FFCO_2$ at urban scales, including their absolute values, spatial gradient, and yearly changes, are substantially different between inventories. The comparisons of the grid scale $FFCO_2$ predictions for selected mega-cities highlight that the grid scale $FFCO_2$ absolute value and temporal change differ largely among bottom-up inventories (Supplementary Fig. S6). The inventories adjusted to the same 2015 value clearly reveal different temporal evolution, with the multi-model spread of up to 1.5-15% during 2016–2018, which are mostly larger than the country-scale analysis (Fig. 5). The KF predictions provide closer multi-inventory agreements in the temporal evolution for many large cities and reduced the multi-inventory spread by 20-80% in 2018. The grid-scale KF prediction thus can provide more consistent temporal dynamics.

The prediction at $2 \times 2°$ resolution reduced global mean errors by 37% relative to predictions at $0.1 \times 0.1°$ resolution. This prediction results in reductions from 5.5 to 3.5% over high emission grids ($FFCO_2$ greater than 0.1 gC/m2/day), while increasing prediction errors locally over several locations (Supplementary Fig. S5). The reduced global mean error suggests that aggregated multi-grid emissions (over 20 grids in this case) provide a smoothed emission trajectory that is suited for the KF prediction. Meanwhile, the increased errors over several locations could reflect the fact that mixed emission sectors that are non-uniformly distributed can complicate the KF prediction. Increasing the prediction resolution to 4x5 degree further reduces the global mean error to 2.7%, while reducing the detailed spatial information. The global mean predictive error is smallest at country scale, 1.9%, which is 65% smaller than the grid-scale ($0.1 \times 0.1°$) prediction.

### Future perspective of emission trajectory

The MEKC regime shifts, which many countries experienced in the past few decades, have important implications for future AQ-GHG mitigation. The Shared Socio-economic Pathways (SSPs) have been developed to describe future scenarios of socioeconomic global change and provide projected GHG and AQ emissions scenarios with different climate policies up to 2100[49,50]. Here we compare four SSPs: SSP1-19, SSP1-26, SSP3-70, and SSP5-85 (from the IPCC's most optimistic scenario to the double $CO_2$ scenarios. See the Supplementary Fig. S7 caption.).

As shown in Supplementary Fig. S7, most of the predicted changes can be well described by the MEKC. Consequently, the overall smooth changes in the emission ratio and the MEKC phase indicate that $FFCO_2$ can be predicted well using the proposal KF prediction for many decades. The Chinese MEKC regime for 2015–2100 in the SSP optimistic scenarios (SSP1-19 and SSP1-26) is Q3 consistent with our analysis after 2014 (Fig. 3), while suggesting the continued current phase after 2018. In the double $CO_2$ scenarios (SPP3-70, SPP5-85), the Q1-Q3 transitions are not predicted until 2100. By contrast, the Chinese transition to Q3 has already occurred by 2018 in our estimate. The $CO_2$/$NO_x$ emission ratios are predicted to increase by 2030 in all the scenarios. While showing similar ratios for recent years (0.28 PgC/TgN in 2015 in SSPs and 0.31 PgC/TgN in 2018 in our analysis), the SSPs predicts maximum emission ratio values of 0.42–0.43 PgC/TgN in 2030 in the SSPs optimistic scenarios, beyond the upper limit for 2018 in this study, suggesting about 0.1 PgC/TgN ratio increase from 2018. This will be followed by reduced ratios from 2030 through 2100 according to stronger $FFCO_2$ reductions.

Also in India, the predicted emission ratios follow the MEKC. The predicted regimes, Q2 during 2015–2020, followed by Q3, in the optimistic scenarios are inconsistent with our estimates (Q1 from 2005 through 2018), while the doubled $CO_2$ scenarios show that it will take many decades to reach Q2 from Q1. A continued increase in the emission ratio by 0.08–0.12 PgC/TgN from 2015 through 2030 in

the optimistic scenarios suggests a possible rapid pathway to achieve economic development while improving AQ. Meanwhile, the predicted ratio of 0.20–0.24 PgC/TgN in 2030 is still smaller than the 2018 ratio in the US and China. This could reflect differences in both technology level and emission sectoral distributions between the countries.

In the US, the observed 2018 phase, Q3, is predicted for 2015–2020 in the optimistic scenarios, whereas it is predicted to reach Q4 in the latter time period when further AQ improvement becomes difficult. The predicted emission ratio increase till 2030 is inconsistent with the already flattened trends before 2018 in our estimates, which are mainly driven by a slow-down in $NO_x$ emission reductions[51]. These scenarios could already overestimate $NO_x$ reductions (or underestimate $FFCO_2$ reductions). The maximum ratio value of about 0.7 PgC/TgN is twice larger than the present value. To achieve the socio-economic level considered in the optimistic scenarios, substantial socio-economic and technological developments would be clearly required. The double $CO_2$ scenarios with Q2 in early years do not match with the actual change (Q3) before 2018, while implying that only $NO_x$ emission reductions may be achieved with fossil fuel based and energy intensive lifestyles.

## Discussion

The MEKC is an important framework for understanding the co-evolution of AQ and carbon emissions in the context of large-scale macroeconomic growth. Based upon this framework using $FFCO_2$ and $NO_x$, the US, China, and India are different locations along the MEKC trajectory, but also change at very different rates. For example, it is remarkable how quickly China shifted MEKC regimes. Within 5 years from 2010, $NO_x$ emissions started returning to 2005 emissions while CO2 emissions stabilized relative to 2005. Furthermore, our results suggest that these trajectories are not independent. For example, China achieved a $CO_2$/$NO_x$ ratio (0.59 TgC/TgN) in the power sector that was roughly 50% higher than the US in 2005 with an equivalent GDP. This suggests that developing countries can take advantage of technology development to reduce AQ emissions. Under the premise that countries will tend to address short-term AQ needs before long-term carbon mitigation, comparing different countries at equivalent GDP could provide insight into their near-term trajectory.

The prediction of $CO_2$ emissions given $NO_x$ emissions bears this out. Dependent on the regime, prediction errors are less than 2% for both developing and developed countries and 5% up to three years for most cases when the the emission ratio changes are temporally smooth. The higher predictive skills for India relative to the US and China reflect the continued linear increase. This predictability can be especially useful for growing economies such as India, which is grappling with substantial AQ challenges. While current results suggest that India continues on a BAU trend, the results from China hold some promise that this trajectory can change fairly quickly with sufficient political and economic demand.

This information could be useful in looking at near-term scenario development. Current IPCC scenarios largely follow a MEKC. However, some scenarios such as doubled $CO_2$ scenarios for China are too pessimistic given our results. Some scenarios suggest $CO_2$/$NO_x$ ratios 20–30% higher than is currently feasible. Progress towards these higher ratios can be monitored with remote sensing, which can provide near-term information. This information is particularly useful for activities such as the Global Stocktake, which requires near-term, e.g., 5-year, assessments. Current predictive errors could be used to assess and adjust emission scenario "story-lines" at this bidecadal cadence consistent with sectoral evolution. For example, the analysis of the MEKC trajectories would provide important implications into low-carbon strategies which could differ between developed and developing countries[10]. The former should focus

more on how to improve energy efficiency and how to change the emission trajectories rather than their initial carbon-intensive infrastructure, whereas the latter, which are currently expanding their energy infrastructure, may have opportunities to leap-frog and bypass carbon-intensive growth.

The accuracy of these predictions is currently contingent on bottom-up approaches. While our current results indicate that we can narrow discrepancies, structural errors can not be fully mitigated. On the other hand, top-down approaches, which use atmospheric $CO_2$, can provide low-latency information, especially for point-sources[52,53], and with increasing capability for urban-scales[48,54,55]. The formulation developed here could be readily adapted to top-down CO2 approaches where our predictions, for example, could help provide AQ-informed priors. Over larger scales where both the biosphere is important and FFCO2 emissions are uncertain, our approach can help partition net carbon fluxes[56] and support attribution[57]. The MEKC concept is a useful interpretive framework for both bottom-up and top-down approaches. Near-term coevolution of AQ and carbon with these data could be used to partition natural and anthropogenic carbon drivers[16,58] and compliment local-scale atmospheric approaches[22].

Additional AQ measurements, e.g., carbon monoxide, can help discriminate sectoral contributions[59] and could be incorporated in future work. Proxies for activity have become increasingly important for near-term carbon emissions estimates but their availability can differ substantially between regions and spatial resolution of the process[60]. Our approach has the advantage of being both transparent and global. Nevertheless, short-term rapid fluctuations in sectoral distribution, and therefore the emission ratio can lead to reduced predictive skill. Additional constraints from proxy information on sectoral distribution changes and the uncertainty estimation results would be helpful to consider these effects more properly.

In order to avoid dangerous climate change, the remaining carbon budget must be managed over increasingly short time horizons. Meeting those targets requires knowledge of emissions and their expected trajectory. The predictive MEKC framework introduced here is useful to both.

## Methods
### FFCO$_2$ emission inventories
**EDGAR v5.0.** Bottom-up emissions of FFCO$_2$ for 2005–2015 were obtained from the EDGAR version 5 inventories[61]. The gridded emissions at 0.1° × 0.1° resolution is extended to 2016–2018 in this study by applying country-scale emission changes provided by the EDGAR 2020 Report[19], while keeping the spatial distributions from EDGAR v5 2015 inventories.

**ODIAC.** The Open-source Data Inventory for Anthropogenic CO$_2$ (ODIAC) is a global high-resolution gridded emissions data product that distributes CO$_2$ emissions from fossil fuel combustion. The emissions spatial distributions were estimated at a 1 × 1 km spatial resolution using power plant profiles and satellite-observed nighttime lights. We used the year 2019 version of the ODIAC emissions data product[18] gridded at 0.1° × 0.1° resolution. The use of bunker fuels was excluded from the analysis.

**FFDAS.** We used the fossil fuel data assimilation system (FFDAS) version 2 data for 2005–2015[62]. It considers electricity-production, industrial, residential, commercial, and transportation (other than domestic aviation and domestic waterborne) sectors, which are similar to the IPCC 1A fuel consumption category. For FFDAS, emission inventories for 2016–2018 were obtained by linear extrapolation from 2014–2015 and used as a reference point to validate the KF predictions.

**GCP-GridFEDv2019.1.** We used GCP-GridFED (version 2019.1), a gridded fossil emissions dataset that is consistent with the national CO$_2$ emissions reported by the Global Carbon Project (GCP). GCP-GridFEDv2019.1 provides monthly FFCO$_2$ for the period 1959–2018 at a spatial resolution of 0.1° × 0.1°[63]. Estimates are provided separately for oil, coal and natural gas, for mixed international bunker fuels, and for the calculation of limestone during cement production. We used the combined emissions except for bunker fuels.

### Top-down NO$_x$ emissions
An updated version of the Tropospheric Chemistry Reanalysis version 2 (TCR-2)[64] is used to evaluate NO$_x$ emission changes. The reanalysis is produced via the assimilation of multiple satellite measurements of ozone, CO, NO$_2$, HNO$_3$, and SO$_2$. The tropospheric NO$_2$ column retrievals from the QA4ECV version 1.1 level 2 product for OMI[65] were used to constrain NO$_x$ emissions.

The forecast model and assimilation technique used were the Model for Interdisciplinary Research on Climate (MIROC)-chemical atmospheric general circulation model for study of atmospheric environment and radiative forcing (CHASER) and an ensemble Kalman filter technique that optimizes both chemical concentrations of various species and emissions.

The global NO$_x$ emissions estimation is based on a state augmentation technique, which has been employed in our previous studies to quantify the spatial and temporal variability of anthropogenic emissions and their impacts on atmospheric composition[26–28,42,44,45,51]. This approach allows us to reflect temporal and geographical variations in transport and chemical reactions in the emission estimates. The emissions in the state vector are represented by scaling factors for each surface grid cell. Only the combined total emission is optimized in data assimilation, where the ratio of different emission categories within the a priori emissions for each grid point were applied to the estimated emissions after data assimilation to obtain the a posteriori anthropogenic emissions. The quality of the reanalysis fields for 2005–2018 has been evaluated based on comparisons against independent observations for various chemical species[64].

Our reanalysis shows the strong increase of Chinese NO$_x$ emission from 2005 to 2011 and a strong decrease after 2014. Emissions from India continue to increase during 2005–2018. Emissions from US and Europe decreased over the past decade. These decadal emission changes reflect the success of AQ mitigation policies and sectoral shifts associated with changes in economy and trade, with implications for AQ and human health.

While top-down NO$_x$ emissions offer great potential to supplement or improve bottom-up inventories, they also contain large uncertainties associated with errors in chemical transport modeling and assimilated observations. Furthermore, any mislocation in the a priori inventories lead to spatial mismatches with the FFCO$_2$ inventories and make the FFCO$_2$ analysis/prediction inadequate. Further detailed comparisons of spatial and temporal emission patterns will play an essential role in improving the prediction.

### Kalman filter technique
The Kalman filter estimates the state of a discrete-time controlled process governed by a linear stochastic difference equation[66]. The operation of the KF includes prediction and correction steps. In the prediction step, an a priori sate of the vector state, $\hat{x}_k$, and its error covariance, $\hat{P}_k$, at the current time step $k$ is projected from the previous time step $k-1$ based on a linear stochastic difference equation:

$$\hat{x}_k = Ax_{k-1} + Bu_k + w_k, \tag{1}$$

where $A$ is the state-transition matrix and $u$ is an external forcing mediated by $B$. The uncertainty of the a state also evolves with time

based upon the following:

$$\hat{P}_k = AP_{k-1}A^T + Q \qquad (2)$$

where $Q$ is the process noise covariance matrix. The measurement model relates an observation to the state:

$$z_k = H_k x_k + \varepsilon_k \qquad (3)$$

where $z_k$ is the observation, $H_k$ is the observation operator, and $\varepsilon_k$ is measurement noise. The updated state $x_k$ is computed from measurements $z_k$ and the a priori state $\hat{x}_k$ through

$$x_k = \hat{x}_k + K_k(z_k - H\hat{x}_k) \qquad (4)$$

where the Kalman gain, $K$, is computed based upon the forecast error covariance $P_k$ and measurements noise covariance matrix, $R$:

$$K_k = \hat{P}_k H^T \left( H\hat{P}_k H^T + R \right)^{-1} \qquad (5)$$

The error covariance is also updated through the following:

$$P_k = (I - K_k H)\hat{P}_k \qquad (6)$$

For this application, $x_k = E_{CO_2}^k / E_{NO_x}^k$ is the scalar emission ratio at time step $k$, which increments annually. To reduce the impact of short-term variability on the prediction, we applied a weighted moving average to $x_k$ (weights of 0.5 for $k-1$ and $k+1$) that helps reduce noise while keeping signals associated with a MEKC regime transition, which helped to improve the KF predictive skill.

## FFCO$_2$ prediction

The FFCO$_2$ prediction approach is illustrated in Fig. 2 and elaborated as:

1. The emission ratio, $x_k$, at time $k$ is predicted from the previous emission ratio $x_{k-1}$ based on the Kalman filter technique, which is described in the previous section (Fig. 2a).
2. Top-down NOx emissions, $E_{NO_x}^k$, are calculated at time $k$ (Fig. 2b).
3. The FFCO$_2$ emissions, $E_{CO_2}^k$, at time $k$ are computed by $x_k \times E_{NO_x}^k$ (Fig. 2c).
4. The updated emission ratio, $x_k$ will be used in the Kalman filter prediction to compute $x_{k+1}$ (then repeat the steps 2–4).

The prediction algorithm is scale-invariant (i.e., a zero-dimensional model) that can be applied to grid-point, country, and continental scales, at any time-scale (e.g., hourly or annual means). For the historical record the predicted ratio, $x_k$ is updated with "observations" $z_k$ of the emission ratio where both bottom-up CO$_2$ and top-down NO$_x$ emissions are available as illustrated by Fig. 2.

For the predictions, top-down NO$_x$ emissions were first downscaled from 1.1° × 1.1° to 0.1° × 0.1° consistent with the resolution of the bottom-up FFCO$_2$ inventories, assuming that bottom-up FFCO$_2$ inventories represent the correct spatial distribution. Then, the bottom-up FFCO$_2$ inventories and converted top-down NO$_x$ emissions both gridded at 0.1° × 0.1° resolution were used for the KF predictions at various spatial scales including regional and country scales.

## Sectoral attribution

The empirical emission ratio, $x_k$, used to compute FFCO$_2$ emissions does not use sectoral information explicitly. However, the updates to this ratio can be projected back to sectors by leveraging a priori

sectoral distributions used in bottom-up inventories as follows:

$$x_k = \frac{E_{CO_2}^k}{E_{NO_x}^k} = \frac{\sum_i E_{CO_2}^{k,i}}{\sum_i E_{NO_x}^{k,i}} \qquad (7)$$

where $i$ denotes a sector. Each sector emission can in turn be written as

$$E_{CO_2}^{k,i} = EF_{CO_2}^{k,i} A_{k,i} \qquad (8)$$

and

$$E_{NO_x}^{k,i} = EF_{NO_x}^{k,i} A_{k,i} \qquad (9)$$

where $EF^{k,i}$ is the emission factor and $A_{k,i}$ is the activity at time $k$. For a given sector, the emission ratio and the emission factor ratio are equivalent:

$$\frac{E_{CO_2}^{k,i}}{E_{NO_x}^{k,i}} = \frac{EF_{CO_2}^{k,i}}{EF_{NO_x}^{k,i}}. \qquad (10)$$

Consequently, changes in activity have no impact on the sectoral emission ratio. In general, this equivalency is not the case for the regional emission ratio in Eq. (7). Changes in $x_k$ can be driven by changes in emission factors or activity across different sectors. However, when an economy is growing and the relative activity between sectors is stable, then changes in $x_k$ will be more sensitive to changes in emission factors. Furthermore, given the co-emission of CO$_2$ and NO$_x$, we would expect changes in sectoral emission factors to impact $x_k$. At national scales, different sectors are correlated leading to coherent changes in national emission ratios.

For the FFCO$_2$ prediction comparisons between the cases with and without sectoral information, the downscaled top-down NO$_x$ emissions were first decomposed into each sectoral emissions, assuming that the bottom-up EDGAR NO$_x$ inventories have the right sectoral distributions at each 0.1° × 0.1° resolution grid. The sectoral top-down NO$_x$ emissions and bottom-up EDGAR FFCO$_2$ inventories were then aggregated into each country and used for the predictions. Errors in the KF predicted total FFCO$_2$ using sectoral and total emissions, $E_i$ and $E_{tot}$, can be represented as $\sum_{i=1}^{n}(E_{NOx,i} \times \varepsilon_i)$ and $E_{NOx,tot} \times \varepsilon_{tot}$, respectively, where $\varepsilon$ is the KF prediction error of emission ratio. Prediction error of total FFCO$_2$ emissions estimated from total emissions, $E_{NOx,tot} \times \varepsilon_{tot}$, can be smaller than those from sectoral emissions, $\sum_{i=1}^{n}(E_{NOx,i} \times \varepsilon_i)$, when an aggregate total emission ratio has a substantially smoother trajectory and a subsequent smaller KF prediction error than individual sectoral emission ratios.

## Uncertainty estimation

Our approach provides uncertainty information of the predicted FFCO$_2$ in the following three ways:

1. KF predictions against the original bottom-up inventories
2. Multi-inventory spreads of the predicted FFCO$_2$
3. Predicted FFCO$_2$ uncertainty from the KF equations

For (1), the prediction errors were estimated to be less than 2% for the 1st year, 3% for the 2nd year, 5% for the 3rd year, and 8% year for the 4th year on average, with slight differences among the countries (Fig. 7 left panels).

For (2), an ensemble of the KF predictions using multi-inventories is used. The choice of emission inventory affects the representation of MEKC dynamics, associated with uncertainty in inventory input data[43]. With removing systematic differences between the inventories at the beginning of the forecast, the use of multiple-inventory resulted in the

KF forecast spread of 0.5–1% for the 1st year and 2–3% in the 4th year in average for the predictions staring in 2010–2015 (red lines in Fig. 7 right panels). Without initial normalization, i.e., with the systematic differences among the inventories, it ranges typically from 3 to 10% (black lines in Fig. 7 right panels).

For (3), the KF predictions involve an evaluation of predicted FFCO$_2$ uncertainty. The error covariance matrix of CO$_2$/NO$_x$ emission ratio ($P_k$, see "Kalman filter technique" section) was updated based on the KF equations. The predicted uncertainty in the current setting at country scale was typically 2–10% for predictions up to 3 years, with 10% errors for $R$ in Eq. (2) and $Q$ in Eq. (5). The error statistics can be refined in future studies as more uncertainty input data become available from both sector-level bottom-up inventories[14] and top-down estimates (e.g., analysis ensemble spreads[42] and multi-model analysis spreads[67]).

These independent uncertainty estimates are in similar magnitudes and can be regarded as typical uncertainty information at country scale.

## Data availability
The NO$_x$ emission data that support the findings of this study are available in https://doi.org/10.25966/9qgv-fe81.

## Code availability
The code we used for data processing is available upon request to the corresponding authors.

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

## Acknowledgements

We acknowledge the use of data products from the NASA Aura and EOS Terra and Aqua satellite missions. We also acknowledge the free use of the tropospheric $NO_2$ column data from the OMI from http://www.qa4ecv.eu. We acknowledge the support of the National Aeronautics and Space Administration (NASA) Atmospheric Composition: Aura Science Team Program (19-AURAST19-0044), the NASA Earth Science U.S. Participating Investigator program (22-EUSPI22-0005), the NASA Carbon Monitoring System (16-CMS16-0027), and the NASA TROPESS project. Part of this work was conducted at the Jet Propulsion Laboratory, California Institute of Technology, under contract with the National Aeronautics and Space Administration (NASA).

## Author contributions

K.M. and K.B. designed and performed research, and wrote and edited the paper.

## Competing interests

The authors declare no competing interests.
