## [Peer Review File · Nature Communications]

Predictability of fossil fuel CO₂ from air quality emissionsReviewer #1 (Remarks to the Author):

#####

GENERAL COMMENT

Even if the idea of predicting fossil fuel CO2 based on air quality emissions is intriguing, I am not convinced about the proposed approach. In particular, as we all know, we are facing (in the last years) rapid changes in activities, energy consumptions...think about COVID, war in Ukraine, energy crisis, etc... How the proposed methodology can correctly model these events, only based on past trends?

So, in my opinion, extrapolating emissions on previous trends can be wrong, if these sudden changes happen in future.

Also, international annual statistics are more robust than real time emissions to represent what has happened over recent years...so a proper emission estimation should start from activity levels.

Finally, if the authors speak about GHG emissions, the model is surely wrong, since GHG are more influenced by other sources, which have different trends compared to NOx and fossil CO2 ... or they may not have even a trend

#####

SPECIFIC COMMENTS

SECTION Modified EKC for FFCO2 and NOx

You say that:

"As the economy continues to mature, both FFCO2 and AQ emissions decrease where FFCO2 will return to its initial values while AQ emissions will be likely less than the normalized year (Q3)."

I do not agree fully with this...i.e., taking the example of road transport emissions in Europe, NOx has been declining while CO2 is still increasing or stabilising. For me, all these assumptions are sector dependent and country dependent, and cannot be generalised with this model

Also the authors say that "To evaluate the predictability, we implemented a simple Kalman filter (KF) of the CO2/NOx emission ratio (Fig. 2a). The FFCO2 emissions (Fig 2c) are then updated based on the product of the CO2/NOx ratio prediction

Also here...one should note that NOx may not be the main driver to represent CO2 emissions ... from residential sector for example..so not clear how effective is the use of this ratio

SECTION Co-evolution of FFCO2 and NOx emissions

The authors say that: "These changes show the MEKC phase shift from Q1 to Q2. A sectoral analysis based on the EDGAR inventories (Fig. S2) suggests that the strong NOx

emission reductions are driven by power industry sources (about 50%) followed by combustion for manufacturing (about 40%) from 2011 to 2018. For FFCO₂, while power industry has the largest contributions to the total increase, the relative growth rate was the largest for road transportation”

NO_x and CO₂ emissions are not characterised by the same relative contribution of the different sectors to the emissions..so these considerations are not fully valid

PREDICTABILITY

The authors say that: “The country-scale results in Fig. 5 is based on a 3-year prediction window, which is a typical latency for bottom-up inventories.”

This is not true, as now also t-1 GHG traditional emission inventories exist.

#####

Reviewer #2 (Remarks to the Author):

The study proposes a novel framework, the modified Environmental Kuznets Curve (MEKC) owing to economic development, air quality regulation, and climate mitigation. They seek to demonstrate that fossil fuel CO₂ emissions can be inferred from NO_x emissions assimilated using satellite NO₂ data, and update FFCO₂ inventories more quickly. I am strongly supportive of such work that seeks to better integrate GHG and AQ observations and emissions, as climate and air quality are two pressing issues facing society across the world. I do have some methodological concerns that need to be addressed before I can recommend this for publication, which mainly pertain to how the analysis is framed and request for more detail. I believe the authors should have a chance to respond my comments outlined below, which hopefully will strengthen the manuscript.

General Comments

(1) An underlying assumption is that the CO₂/NO_x ratio is slow-evolving relative to activity changes, hence why FFCO₂ can be predicted from assimilation of satellite NO₂ data within a given MEKC. While this could be true in the aggregate at the national scale, it is hard to explain how this is consistent with the literature for individual sectors. For example, AQ regulations have really decreased the NO_x/CO₂ emissions ratio from US power plants (-10%/y) and transportation (-5 to -10%/y), which are two key NO_x emitting sectors. A more careful description of the strength and limitations of leveraging the CO₂/NO_x ratio is needed throughout, especially of how CO₂/NO_x trends of individual emission sectors (Fig.4) affect emission ratios when aggregated to regional to national scales. A caveat is provided in the discussion, “The accuracy of these predictions is currently contingent on bottom-up approaches. While our current results indicate that we can narrow discrepancies, structural errors can not be fully mitigated.” The manuscript would be strengthened if this were addressed earlier on in the manuscript and more directly, such as in the CO₂/NO_x emission ratio section.

de Gouw, J. A., et al. (2014). "Reduced emissions of CO₂, NO_x, and SO₂ from U.S. power plants owing to switch from coal to natural gas with combined cycle technology." *Earth's Future* 2: 75-82.

Yu, K. A., et al. (2021). "Evaluation of Nitrogen Oxide Emission Inventories and Trends for On-Road Gasoline and Diesel Vehicles." *Environmental Science & Technology*.

(2) Building on Comment #1, it is hard to follow the Sectoral Analysis section since it is

not clear in the methods section how the top-down NO_x inventory has sectoral information, which seems like it would be needed to derive CO₂ emissions from the FFCO₂:NO_x ratio. A better description of the NO_x downscaling approach is needed, and a clearer delineation between where top-down or bottom-up methods are utilized to inform both NO_x and FFCO₂ from sectors to regional to country scales. Also, it is unclear in the FFCO₂ prediction section whether the bottom-up equations are just for CO₂ or also NO_x.

(3) I have concerns with the oversimplification of GHG as CO₂ and AQ as NO_x. Obviously, there are other important GHGs (e.g., CH₄) and VOCs and PM_{2.5} that contribute to AQ. Why not just be more specific and reference as a FFCO₂-NO_x framework? I don't think that will minimize the significance of the work or attempts to marry GHG-AQ inventories. The over-simplification undervalues the complexity of dealing with the climate and AQ issues at hand.

(4) The authors have spent a good amount of effort on uncertainty estimation. Can this uncertainty estimation also be reflected in Figure 3, which is a key figure for the paper? Also, why are the Q2 sections so brief in Figure 3b for the US and China?

(5) If we can predict FFCO₂ within 2% (one-year lag) and 10% (four-year lag) from NO₂, why is there a need for a global carbon monitoring system in future? I don't think this is what the authors intended, but it might help to elaborate further in the discussion on how having satellite CO₂ observations within the KF framework could affect the analysis in the future.

Author's comments in reply to the anonymous referee for "Predictability of fossil fuel CO₂ from air quality emissions" by Miyazaki and Bowman

Reply to Referee #1

We would like to thank the referee for the helpful comments. We have revised the manuscript according to the comments, and hope that the revised version is now suitable for publication. Below are the referee comments in blue italics with our replies in normal font.

GENERAL COMMENT

Even if the idea of predicting fossil fuel CO₂ based on air quality emissions is intriguing, I am not convinced about the proposed approach. In particular, as we all know, we are facing (in the last years) rapid changes in activities, energy consumptions...think about COVID, war in Ukraine, energy crisis, etc... How the proposed methodology can correctly model these events, only based on past trends? So, in my opinion, extrapolating emissions on previous trends can be wrong, if these sudden changes happen in future.

We agree that the predictive scheme depends on the underlying dynamics, e.g., we can't expect a weather forecast to be equally accurate everywhere. Overall, however, we have shown that predicting the *ratio* ECO_2/ENO_x is reasonably accurate both historically and under future IPCC scenarios. Under more sudden changes, for example, when an economic crisis occurs, overall activity will plummet across multiple sectors at the same time, especially across large scales where multi-sector activities, such as transportation and manufacturing, are strongly coupled. In this case, the *emission ratio* will remain the same even as the activity drops. A good example is the rapid reduction in NO_x emissions, for instance, during the COVID lockdowns, which was captured by our chemical data assimilation system (Miyazaki et al., Science Advances, 2021). The reduction in *activity* is captured by the NO_x emissions, which can be propagated into FFCO₂ emission reductions through the emission ratio. The accuracy of that estimate will depend on the relative contribution of different sectors, which drives the ratio, to remain substantially the same. If that contribution changes, e.g., only the transportation sector changes, the ratio prediction will incur an additional error. We have detailed in the manuscript cases where this can occur, e.g., 2007-2008 financial crisis.

To summarize, rapid economic changes, e.g., COVID, can rapidly affect both activity and emission ratios, but for different reasons. Our methodology does allow us to capture rapid changes in activity through

top-down NO_x emissions but will incur an error in the FFCO₂ prediction if the emission ratio changes rapidly.

The focus of our manuscript, however, is on larger-scale changes related to the MEKC. In that case, when new regulations or technological development occurs, changes in sectoral distributions and in emission ratios can lead to poorer predictive skills, especially when the MEKC regime transitions from one to the next. Nevertheless, implementing new regulations and technological development usually happen gradually, and resultant gradual changes in emission ratio can provide FFCO₂ predictive skills.

We agree with the reviewer that further discussion about FFCO₂ predictions under rapidly changing events would be helpful. While the manuscript demonstrated that the general proposed approach is valid for most cases at country scales, we seriously considered the limitation of our approach to rapidly changing events and discussed them carefully in the manuscript as follows:

In “Modified EKC for FFCO₂ and NO_x”:

“The KF prediction exploits the relatively gradual changes in emission ratio with a given MEKC regime. **Even when the activity changes rapidly, multi-sector activities can be strongly linked at country scales and can provide robust KF predictions.** However, during the transition from one regime to the next, we would expect poorer performance.”

In “Predictability”:

“The exception is India in 2007 and 2010 where the 1-year lag error exceeded 3%. Likewise, US prediction overestimated FFCO₂ reductions from 2006 to 2007 related to the economic crisis (Fig. 5c). Short-term fluctuations in GDP are not well-modeled in the MEKC and are reflected in the skill of the prediction. In general, however, these errors are smaller than the spread in current emission inventories (6-7 %). **Rapid changes in emissions are often driven by changes in activity that are well-informed by satellite-constrained NO_x emissions, e.g., COVID-19 lockdowns (Miyazaki et al, 2021, Laughner et al, 2021) and cut across multiple sectors. To the extent that the relative sectoral impacts are the same, the FFCO₂ will be robust. Over longer time scales, the predictive skill suggests that emission ratios tend to change more slowly than activity.**”

In “Predictability”:

“However, the approach does not reproduce the dynamics when the emission ratio changes rapidly, such as in India in 2010 and the USA in 2007 during the economic crisis. These anomalies could not impact all sectors equally, which leads to a change in the aggregate emission ratio, and therefore degrades the

prediction skill, especially when predicting FFCO₂ at small scales where the relative sectoral distribution can change substantially, e.g., transportation relative to power production. At country scales, however, multi-sector activities are highly coupled and therefore provide robust predictive skill for many cases, as discussed later.”

In Discussion:

“Nevertheless, short-term rapid fluctuations in sectoral distribution, and therefore the emission ratio can lead to reduced predictive skill. Additional constraints from proxy information on sectoral distribution changes and the uncertainty estimation results would be helpful to consider these effects more properly.”

Also, international annual statistics are more robust than real time emissions to represent what has happened over recent years...so a proper emission estimation should start from activity levels.

Yes, we agree that international annual statistics are the most robust source of information. That’s why we used bottom-up inventories, which are informed by international statistics, to train the Kalman prediction algorithm. We combine sectoral distributions, international annual energy statistics with low latency information on NO_x emissions, which is used as a proxy of activity data, to predict the trajectory of the FFCO₂/NO_x emission ratio and estimate FFCO₂. Independent constraints on activity from satellite NO₂ observations are particularly important for global analysis as emission factors and activity data used in bottom-up inventories can still have significant regionally-dependent uncertainties for developing countries. This is clearly described in the revised manuscript as follows:

In “Modified EKC for FFCO₂ and NO_x”:

“While utilizing sectoral distributions from bottom-up inventories informed by international statistics, our approach exploits both the rapid update of NO_x emissions enabled by satellite assimilation and the gradual changes in technology and regulation (i.e., emission ratio).”

Finally, if the authors speak about GHG emissions, the model is surely wrong, since GHG are more influenced by other sources, which have different trends compared to NO_x and fossil CO₂ ... or they may not have even a trend

We agree that different GHGs, such as CH₄, tropospheric ozone, and N₂O, are driven by different processes than FFCO₂ and therefore would require different predictors. We note that other AQ species, such as ammonia, could be highly relevant for GHG sectors, e.g., livestock or agriculture. Nevertheless, in the presented study, we demonstrated this methodology against the important GHG (FFCO₂) and one

of the most important and measurable AQ (NO_x) only. We can think of our manuscript as treating a particular instance of GHG and AQ synergy rather than a comprehensive treatment of all GHG and AQ. Based upon a comment by another reviewer, the label of the MEKC schematic figures has been changed to “AQ (NO_x)” and “GHG (FFCO₂)” in the revised manuscript to avoid any confusion and overstatement. The following sentence has been added to clarify this point:

In “Modified EKC for FFCO₂ and NO_x”:

“Here we use FFCO₂ and NO_x as one of the most measurable GHG and AQ species and good proxies of various co-emission sources to describe the MEKC trajectory. MEKC applications to other forms of co-emitted GHG-AQ species would provide different trajectories and unique insights into the economy and emission relationship. Nevertheless, its concept provides a generalized framework to describe the emission trajectory dynamics.”

SPECIFIC COMMENTS

SECTION Modified EKC for FFCO₂ and NO_x

You say that:

“As the economy continues to mature, both FFCO₂ and AQ emissions decrease where FFCO₂ will return to its initial values while AQ emissions will be likely less than the normalized year (Q3).”

I do not agree fully with this...i.e., taking the example of road transport emissions in Europe, NO_x has been declining while CO₂ is still increasing or stabilising. For me, all these assumptions are sector dependent and country dependent, and cannot be generalised with this model

Thank you for this comment. We agree that the prediction is both country and sector dependent. That doesn't mean, however, that they cannot be generalized. The prediction is trained separately so that the trajectory is unique to each country, e.g., Fig 3. While we have demonstrated that the MEKC concept is generally robust, it will not work everywhere and for all sectors equally well.

As indicated by the reviewer and shown in Fig. A1, the NO_x emission declines while CO₂ increases for European road transportation. It has been reported that CO₂ emissions from heavy-duty vehicles have increased every year since 2014 and that the efficiency gains that have been achieved in vehicles and transport operations have been outpaced by the growing demand for freight transport. This has led to continued growth in emissions.

Fig. A1: NO_x and FFCO₂ emission changes for road transportation over Western European countries.

While the MEKC provides a useful framework to interpret GHG-AQ co-evolution, the filtering algorithm is not explicitly dependent on it. Consequently, the smooth emission trajectory would still provide a good predictive skill based on the KF prediction, even if it does not follow the MEKC trajectory. To highlight possible exceptions and describe the limitation of the MEKC concept, the following sentences have been added to the revised manuscript:

In “Sectoral analysis”:

“The MEKC framework is robust for interpreting GHG-AQ co-evolution when integrated over coupled sectors typical of countries scales. However, individual sectors may deviate from the MEKC. For example, FFCO₂ from European transportation has increased since 2013 while NO_x emissions decline due to the growing demand for freight transport and the effective AQ regulation for heavy-duty vehicles. That sector change is more reflective of Q2 even though Western Europe as a whole is in Q3 where both CO₂ and NO_x emissions have reduced. At regional scales, the ratio of aggregated sectoral CO₂ emissions to aggregated sectoral NO_x emissions is not equal to emission ratios aggregated over sectors (see Methods). Consequently, aggregated over all the major sectors, countries such as US, India, and China, and western Europe follow the MEKC regimes, but individual sector emission ratio trajectories may have distinctly different trends.”

Also the authors say that “To evaluate the predictability, we implemented a simple Kalman filter (KF) of the CO₂/NO_x emission ratio (Fig. 2a). The FFCO₂ emissions (Fig 2c) are then updated based on the product of the CO₂/NO_x ratio prediction

Also here...one should note that NOx may not be the main driver to represent CO2 emissions ... from residential sector for example..so not clear how effective is the use of this ratio

Yes, there are sectors with low co-emissions of NO_x and FFCO₂, such as the residential sector, which if treated in isolation would lead to poor predictive skills. This limitation would be particularly true for high-resolution NO_x emissions estimates to predict FFCO₂ exclusively over residential sectors. Nevertheless, at larger scales (e.g., country scales), sectoral emissions, such as residential emissions and power energy, are coupled so that the aggregated emission ratio still accounts for the majority of these emissions. Therefore, the combined country-total emission still follows the MEKF regimes, as presented in this study. Please also see our reply to the relevant comments above.

SECTION Co-evolution of FFCO₂ and NO_x emissions

The authors say that: “These changes show the MEKC phase shift from Q1 to Q2. A sectoral analysis based on the EDGAR inventories (Fig. S2) suggests that the strong NO_x emission reductions are driven by power industry sources (about 50%) followed by combustion for manufacturing (about 40%) from 2011 to 2018. For FFCO₂, while power industry has the largest contributions to the total increase, the relative growth rate was the largest for road transportation”

NO_x and CO₂ emissions are not characterised by the same relative contribution of the different sectors to the emissions...so these considerations are not fully valid

When the NO_x and CO₂ emissions are characterized by the same relative contribution, then the GHG-AQ emission ratio will not change. In the example mentioned above, we are explaining why the aggregated emission ratio changed by attributing that change to specific sectoral shifts, as illustrated in Fig. S3 in the revised manuscript (please see below). This doesn't invalidate the results, but provides insight into how these emission dynamics fit within a MEKC framework—and how they are reflected in our prediction performance. The EDGAR sectoral information provides important supplementary information in this study to understand the influences of sectoral distribution changes. To more clearly state them, the relevant sentences have been rewritten as follows:

In “Sectoral analysis”:

"The comparisons also highlight that the impact of the sectoral shifts informed by bottom-up inventories is well reflected as a whole in changes in an aggregated country total emission ratio. An aggregate emission ratio usually shows smooth trajectories (Fig. S3). Since a KF prediction error of total emissions can be represented as a sum of sectoral emission ratio prediction errors (see the

methods section), a smooth trajectory of an aggregate emission ratio is more amenable to KF predictions."

Fig. S3: Relationships between Gross domestic product (GDP, in USD) and CO₂/NO_x emission ratio for each emission sector separately: power industry (POW), combustion for manufacturing (COM), road transportation (TRA), and for total emissions (All). The results are shown for China, India, and USA.

PREDICTABILITY

The authors say that: "The country-scale results in Fig. 5 is based on a 3-year prediction window, which is a typical latency for bottom-up inventories."

This is not true, as now also t-1 GHG traditional emission inventories exist.

Thank you for the comments. While the typical latency of the traditional grid-based and sectoral emission inventories is three years (such as EDGAR and REAS), we agree with the reviewer that there has been an increasing attempt to reduce its latency. Even in this case, our predictions, with a minimum latency of a few weeks, can provide a much faster and accurate FFCO₂ predictions. Accordingly, the sentence has been rewritten as:

"The country-scale results in Fig. 5 are based on a 3-year prediction window, which is a typical latency for comprehensive bottom-up inventories while there has been an increasing attempt to reduce its latency up to a year using relatively simplified settings."

Reply to Referee #2

We appreciate this positive assessment and would like to thank the referee for the helpful comments. We have revised the manuscript according to the comments, and hope that the revised version is now suitable for publication. Below are the referee comments in blue italics with our replies in normal font.

The study proposes a novel framework, the modified Environmental Kuznets Curve (MEKC) owing to economic development, air quality regulation, and climate mitigation. They seek to demonstrate that fossil fuel CO₂ emissions can be inferred from NO_x emissions assimilated using satellite NO₂ data, and update FFCO₂ inventories more quickly. I am strongly supportive of such work that seeks to better integrate GHG and AQ observations and emissions, as climate and air quality are two pressing issues facing society across the world. I do have some methodological concerns that need to be addressed before I can recommend this for publication, which mainly pertain to how the analysis is framed and request for more detail. I believe the authors should have a chance to respond my comments outlined below, which hopefully will strengthen the manuscript.

We appreciate the positive comments. The manuscript has been improved substantially corresponding to the reviewer's comments.

General Comments

(1) An underlying assumption is that the CO₂/NO_x ratio is slow-evolving relative to activity changes, hence why FFCO₂ can be predicted from assimilation of satellite NO₂ data within a given MEKC. While this could be true in the aggregate at the national scale, it is hard to explain how this is consistent with the literature for individual sectors. For example, AQ regulations have really decreased the NO_x/CO₂ emissions ratio from US power plants (-10%/y) and transportation (-5 to -10%/y), which are two key NO_x emitting sectors. A more careful description of the strength and limitations of leveraging the CO₂/NO_x ratio is needed throughout, especially of how CO₂/NO_x trends of individual emission sectors (Fig.4) affect emission ratios when aggregated to regional to national scales. A caveat is provided in the discussion, "The accuracy of these predictions is currently contingent on bottom-up approaches. While our current results indicate that we can narrow discrepancies, structural errors can not be fully mitigated." The manuscript would be strengthened if this were addressed earlier on in the manuscript and more directly, such as in the CO₂/NO_x emission ratio section.

de Gouw, J. A., et al. (2014). "Reduced emissions of CO₂, NO_x, and SO₂ from U.S. power plants owing to switch from coal to natural gas with combined cycle technology." Earth's Future 2: 75-82.

Yu, K. A., et al. (2021). "Evaluation of Nitrogen Oxide Emission Inventories and Trends for On-Road Gasoline and Diesel Vehicles." Environmental Science & Technology.

A more careful description of the strength and limitations of leveraging the CO₂/NO_x ratio is needed throughout, especially of how CO₂/NO_x trends of individual emission sectors (Fig.4) affect emission ratios when aggregated to regional to national scales.

We agree that the relationship between regional emission ratios and the emission ratios of individual sectors can be subtle and at times counter-intuitive. The assumption that emission factors are slow relative to activity is valid within a MEKC regime but breaks down when transitioning from one regime to another. We discuss this point in detail in the Predictability section. At country scales, different sectors' activity, such as residential emissions and power energy, can be coupled in many cases, and therefore the combined country-total emission still follows the MEKC regimes. As shown in Fig. S3 in the revised manuscript, an aggregate emission ratio generally shows a smoother trajectory than individual sector emission ratios. Since a KF prediction error of total emissions can be represented as a sum of sectoral emission ratio prediction errors, a smoother trajectory of an aggregate emission ratio is considered to be more suitable for total emission predictions. This would reduce the influences of more complicated patterns of each sector's emission trajectory.

Looking at Fig. S1, we can see a transition in the US emission ratio from about 1999 to 2012. We also see the CO₂/NO_x emissions for the US increase from about 0.28 to 0.35 over a 5-year period from 2005. These results are qualitatively consistent with de Gouw et al, but aren't as rapid. That may be attributable to different overlap periods from de Gouw (1995-2012) and our analysis (focused from 2005-2018) where that transition is already well underway. There's also the difference between the inventories themselves and the CEMS data.

To address your concern, we reorganized the Methods Section and rewrote the sectoral attribution section. In this new section, we explicitly relate the regional emission ratio to the sectoral emissions. These sectoral emissions are in turn related to emission factors and activity. From this formulation, we hope it is clear that the regional emission ratio is not simply the sum of sectoral emission ratios. However, this formulation does provide a way to compute the regional emission ratios from individual sector emissions of NO_x and CO₂. So, different estimates for some sectors could be related to our estimates (given information from other sectors).

Furthermore, we have added text to help clarify the limitations of our approach:

In “Predictability”:

“However, the approach does not reproduce the dynamics when the emission ratio changes rapidly, such as in India in 2010 and the USA in 2007 during the economic crisis. These anomalies could not impact all sectors equally, which leads to a change in the aggregate emission ratio, and therefore degrades the prediction skill, especially when predicting FFCO₂ at small scales where the relative sectoral distribution can change substantially, e.g., transportation relative to power production. At country scales, however, multi-sector activities are highly coupled and therefore provide robust predictive still for many cases, as discussed later.”

In “Modified EKC for FFCO₂ and NO_x”:

“Even when the activity changes rapidly, multi-sector activities can be strongly linked at country scales and can provide robust KF predictions. However, during the transition from one regime to the next, we would expect poorer performance.”

In “Sectoral analysis”:

“The emission ratios show distinctly different patterns among sectors (Fig. 4). For instance, in China and the US, the emission ratio of the power industry emission increased due to the AQ regulations and the increased use of natural gas (Gouw et al., 2014). Also, in the US, total on-road NO_x emissions declined after 2004 when heavy-duty diesel NO_x emission controls started (Katelyn et al., 2021), which increased the emission ratio.”

In “Sectoral analysis”:

"The comparisons also highlight that the impact of the sectoral shifts informed by bottom-up inventories is well reflected as a whole in changes in an aggregated country total emission ratio. An aggregate emission ratio usually shows smoother trajectories than individual sectoral emission ratios (Fig. S3). Since a KF prediction error of total emissions can be represented as a sum of sectoral emission ratio prediction errors (see the methods section), a smoother trajectory of an aggregate emission ratio is considered to be more suitable for the KF predictions."

In “Sectoral attribution”:

“Errors in the KF predicted total FFCO₂ using sectoral and total emissions, $\$E_i\$$ and $\$E_{\text{tot}}\$$, can be represented as $\$¥\sum_{i=1}^n ¥\left(E_{\text{NOx},i} ¥\times ¥\epsilon_i ¥\right)\$$ and $\$E_{\text{NOx,tot}} ¥\times ¥\epsilon_{\text{tot}}\$$, respectively, where $\$¥\epsilon\$$ is the KF prediction error of emission ratio.

Prediction error of total FFCO₂ emissions estimated from total emissions, $E_{\text{NOx,tot}} \times \epsilon_{\text{tot}}$, can be smaller than those from sectoral emissions, $\sum_{i=1}^n \left(E_{\text{NOx},i} \times \epsilon_i \right)$, when an aggregate total emission ratio has a substantially smoother trajectory and a subsequent smaller KF prediction error than individual sectoral emission ratios.”

(2) Building on Comment #1, it is hard to follow the Sectoral Analysis section since it is not clear in the methods section how the top-down NOx inventory has sectoral information, which seems like it would be needed to derive CO2 emissions from the FFCO2:NOx ratio. A better description of the NOx downscaling approach is needed, and a clearer delineation between where top-down or bottom-up methods are utilized to inform both NOx and FFCO2 from sectors to regional to country scales. Also, it is unclear in the FFCO2 prediction section whether the bottom-up equations are just for CO2 or also NOx.

The methods section “FFCO₂ prediction”, “Kalman Filter”, and “Sectoral Analysis” have been substantially revised to more clearly describe the downscaling approach, sectoral decomposition approach, and the use of top-down and bottom-up emissions, while extending the bottom-up equations to the evaluation of KF-predicted emission errors.

(3) I have concerns with the oversimplification of GHG as CO2 and AQ as NOx. Obviously, there are other important GHGs (e.g., CH4) and VOCs and PM2.5 that contribute to AQ. Why not just be more specific and reference as a FFCO2-NOx framework? I don't think that will minimize the significance of the work or attempts to marry GHG-AQ inventories. The over-simplification undervalues the complexity of dealing with the climate and AQ issues at hand.

We used FFCO₂ and NOx as a specific example and an initial attempt of the GHG-AQ relationship described within the MEKC concept, since they are the most measurable species and good proxies of various co-emission sources. While other GHG and AQ species would represent different trajectory patterns in the MEKC dynamics, unique knowledge of the economy and emission relationship could be obtained from MEKC applications to other GHG-AQ species. This is more carefully described in the revised manuscript. Considering its possible broader contributions, we still call the framework as the GHG-AQ relationship in the paper, because it is a general concept and is applicable to other species. Meanwhile, the following sentences have been added to the revised manuscript to clarify its limited application in this study:

“Here we use FFCO₂ and NOx as one of the most measurable GHG and AQ species and good proxies of various co-emission sources to describe the MEKC trajectory. MEKC applications to other forms of

co-emitted GHG-AQ species would provide different trajectories and unique insights into the economy and emission relationship. Nevertheless, its concept provides a generalized framework to describe the emission trajectory dynamics.”

The application to FFCO₂ and NO_x in this study is more clearly mentioned throughout the revised manuscript.

(4) The authors have spent a good amount of effort on uncertainty estimation. Can this uncertainty estimation also be reflected in Figure 3, which is a key figure for the paper? Also, why are the Q2 sections so brief in Figure 3b for the US and China?

Fig 3 represents the evolutions of emissions and emission ratios from the bottom-up and top-down inventories. We used the estimated multi-inventory spread of four FFCO₂ inventories to add the error bars to the emission ratio trajectories. Note that, because the results shown are not the KF prediction results, the KF prediction uncertainty cannot be added to this figure.

The short Q2 prediction in China is due to the rapid AQ regulation in 2012 followed by the AQ-carbon regulation in 2014. In the United States, the rapid decline in FFCO₂ in 2012, which was caused by the combustion for manufacturing emission reduction in the EDGAR inventory, leading to the transient Q2 phase.

(5) If we can predict FFCO₂ within 2% (one-year lag) and 10% (four-year lag) from NO₂, why is there a need for a global carbon monitoring system in future? I don't think this is what the authors intended, but it might help to elaborate further in the discussion on how having satellite CO₂ observations within the KF framework could affect the analysis in the future.

Point source CO₂ emissions could be predicted using our Kalman filter concept and then updated with direct CO₂ measurement from a satellite such as PRISMA or eventually CarbonMapper. Our approach doesn't remove the need for better CO₂, especially at finer scales but provides complementary information, especially at scales amenable to a MEKC. This is particularly true for larger area fluxes where the biospheric contribution is substantial and the FFCO₂ uncertainty is non-negligible. To describe this, the following sentences have been added.

“On the other hand, top-down approaches, which use atmospheric CO₂, can provide low-latency information, especially for point-sources (Nassar et al, 2022, Cusworth et al, 2021a), and with increasing

capability for urban-scales (Kiel et al, 2021, Mueller et al, 2021, Yang et al, 2020). The formulation developed here could be readily adapted to top-down CO₂ approaches where our predictions, for example, could help provide AQ-informed priors. Over larger scales where both the biosphere is important and FFCO₂ emissions are uncertain, our approach can help partition net carbon fluxes (Yin et al, 2019) and support attribution (Cusworth et al, 2021b). The MEKC concept is a useful interpretive framework for both bottom-up and top-down approaches”

The key science points that have been discovered need to be brought out more in the abstract and the conclusions.

The conclusions have been revised to bring out more key science results. We were unable to add to the abstract because of the word limit. We tried to balance the need to introduce the concept while also capturing key quantitative science points. We hope this will address most of your concerns.

Reviewer #1 (Remarks to the Author):

I think the authors did a good job in replying to my comments. I think now the paper can be accepted for publication.

Reviewer #2 (Remarks to the Author):

The authors have sufficiently addressed my concerns, and I recommend for publication.